# Research on Air-Conditioning Cooling Load Correction and Its Application Based on Clustering and LSTM Algorithm

**Honglian Li \*, Li Shang, Chengwang Li and Jiaxiang Lei**

School of Information and Control Engineering, Xi'an University of Architecture and Technology, Xi'an 710055, China; 15735102971@163.com (L.S.)
**\*** Correspondence: lihonglian_lhl@163.com

**Abstract:** Climate change and urban heat island effects affect the energy consumption of buildings in urban heat islands. In order to meet the requirements of engineering applications for detailed daily design parameters for air conditioning, the 15-year summer meteorological data for Beijing and Shanghai and the corresponding average heat island intensity data were analyzed. Using the CRITIC objective weighting method and K-means clustering analysis, the hourly change coefficient, $\beta$, of dry bulb temperature was calculated, and the LSTM algorithm was used to predict the changing trends in $\beta$. Finally, the air conditioning load model for a hospital was established using DeST (version DeST3.0 1.0.2107.14 20220712) software, and the air conditioning cooling load in summer was calculated and predicted. The results show that, compared with the original design days, regional differences in the new design days are more obvious, the maximum temperature and time have changed, and the design days parameters are more consistent with the local meteorological conditions. Design day temperatures in Shanghai are expected to continue rising for some time to come, while those in Beijing are expected to gradually return to previous levels. Among hospital buildings, the cooling load of outpatient buildings in Beijing and Shanghai will decrease by 0.69% and increase by 12.61% and by 12.12% and 15.51%, respectively, under the influence of the heat island effect. It is predicted to decrease by 1.35% and increase by 29.75%, respectively, in future. The cooling load of inpatient buildings in Beijing and Shanghai increased by 0.27% and 6.71%, respectively, and increased by 7.13% and 8.09%, respectively, under the influence of the heat island effect, and is predicted to decrease by 0.93% and increase by 16.07%, respectively, in future.

**Keywords:** urban heat island effect; cluster analysis; CRITIC objective weighting method; LSTM prediction model; air conditioning load parameter modification; hospital building

## 1. Introduction

Climate change is an indisputable fact. The fifth report of the Intergovernmental Panel on Climate Change (IPCC) pointed out that the global surface temperature increased from 1880 to 2012, with the global average temperature increasing by 0.85 °C. Based on the largest temporal dataset [1], the average temperature difference between 1885 and 1900 and between 2003 and 2013 was 0.78 °C. The rate of warming in China from the 1970s to the early 2000s was greater in the last 100 years than in any previous decade [2]. Climate change has local characteristics. Local climate change is the response to climate change on the regional or urban scale and is closely associated with urbanization. The heat island effect is one of the most significant effects of urbanization on urban temperature [3]. The urban heat island effect is one of the typical characteristics of an urban climate, a concept put forward in the early 20th century by British climatologists. The urban heat island effect refers to the phenomenon that the temperature in a city is higher than in the surrounding areas. Under the influence of global climate change, the urban heat island effect is increasingly intensified, seriously affecting the health, thermal comfort, and other urban ecological functions of urban residents, and it is an important factor restricting the

healthy development of cities [4]. The urban area warming phenomenon caused by the urban heat island effect can lead to a change in the urban thermal environment and increase the possibility of extremely high temperatures in urban areas, thus increasing climate risk and affecting the quality of the ecological environment, human comfort, and health [5]. At present, research on urban heat islands has matured. As shown in Table 1, many scholars at home and abroad have conducted high-quality research in related fields using different methods and parameters.

**Table 1.** Related research.

| Research Idea | Research Content | Relevant Data | Literature |
|---|---|---|---|
| Fixed or moving observations from the ground (meteorological stations and meteorological monitoring equipment). | Urban albedo | Area of the city | Oka [6] |
| | Cloud cover and wind speed | Twenty years of data from 40 weather stations in Melbourne. | Morris [7] |
| | Temperature and wind speed | Barrow meteorological data. | Hinkle [8] |
| | Wind | Data from 77 weather stations in London. | Giridhar an [9] |
| | Temperature | Meteorological data for Athens from 1970 to 2004. | Found [10] |
| Remote sensing data inversion (remote sensing data, infrared thermal data) | Surface radiant temperature | Remote sensing data for U.S. coastal areas | Roth [11] |
| | Single window algorithm | Landsat TM data for Egypt and Israel | Qin [12] |
| | Surface radiant temperature | NOAA thermal infrared data of four coastal cities in Greece | Author [13] |
| | Gaussian approximation method to quantify surface temperature. | Remote sensing data for 8 cities | Tran [14] |
| | Building density and roofing materials. | Landsat-7 EMT+ data and SPOT imaging data from Seoul, Korea | Bhang [15] |
| | Level of commercialization | Landsat-7 EMT+ satellite image data from Delhi, India | Mallick [16] |
| Numerical simulation method (the data model) | Three-dimensional mesoscale model | Meteorological and geographic data for Brisbane | Khan [17] |
| | Three-dimensional flow model | Emissivity, albedo, and thermal inertia data for London city | Atkinson [18] |

Miftah [19] pointed out that the changes in underlying surface properties and anthropogenic heat emissions are the two main factors causing the urban heat island phenomenon. Liu [20] pointed out that a large number of impervious surfaces, including asphalt and cement, replaced the original natural surface. At the same time, the daily heat emissions for urban residents surged, leading to an increase in urban heat emissions and increasing the urban heat island effect. Zhan [21] pointed out that unreasonable spatial urban layouts will further aggravate the negative effects of climate change. Moon [22] showed that a series of environmental and ecological problems caused by urban heat islands have affected the further development of cities and changed the ecological environment of human settlements. Based on statistical data [23], the global heat island effect can lead to an average reduction in heating energy consumption of buildings by 18.7% and an average increase in cooling energy consumption of 19.0%, which has an impact on the total building energy consumption. Therefore, it is necessary to study the energy consumption of buildings under influence of the urban heat island effect.

The influence of the urban heat island effect on the energy consumption of urban buildings, especially on the energy consumption of building air conditioning systems, has been the focus of researchers. Although research on building occupants has proliferated over the past decade, buildings are still designed and operated based on outdated and simplistic occupant assumptions that are increasingly proven to be wrong [24]. According to statistics [25], in China, building energy consumption accounts for nearly one-third of the total energy consumption. In 2018, China consumed 2.147 billion tons of equivalent coal in

the whole life cycle of buildings, of which 1.1 billion tons of equivalent coal was consumed in building material production, accounting for 51.3 percent of the total energy consumption. The figures for the construction phase and operation phase were 47 million TCE (2.2%) and 1 billion TCE (46.6%), respectively. Perez Lombard et al. [26] analyzed the existing information on building energy consumption, especially the information related to high-voltage air conditioning systems. On the basis of a detailed investigation of the availability of building information, considering the commercial buildings between different countries, the situation in offices can elicit more in-depth analysis. Feng et al. [27] found that due to the different opening times of air conditioning, the energy consumption of different buildings varies greatly. Through building energy simulation, the behavior of residents is classified. These typical behavior patterns of residents can be used as the background of community building energy profile and the evaluation standard of building energy saving technology. Li et al. [28] measured the energy consumption of refrigeration in a residential building in Beijing, China, in summer. The results show that the running time of air conditioning has a significant effect on building energy consumption. Based on the Gini coefficient of air conditioning energy consumption, the operation mode of air conditioning is regarded as the most important factor affecting the energy consumption of residential buildings. At the same time, it is found that China's residential building refrigeration energy consumption has great potential for growth and decline. Santamaria et al. [29] conducted a comparative analysis of different building cases, and the results showed that under the urban heat island effect, the cooling load of buildings located in the city center was 13% higher than that of buildings located in the suburbs. At the same time, Santamaria also points out that the instantaneous power consumption of buildings increases by 4.6% and the total power consumption increases by 8.5% when the temperature increases by 1 °C [30]. Temperature rise has a significant impact on building energy consumption in different climate zones in China. By using the "Morphing" deformation method [31], the meteorological parameters, such as dry bulb temperature, relative humidity, and solar radiation, in the location of the building are revised, and Energy Plus is applied to simulate the office building to study the influence of the urban heat island effect on building energy consumption. In general, the increase in urban heat island intensity and the decrease in relative humidity and wind speed lead to a decrease in the heating load and an increase in the cooling load of residential buildings [32]. The average annual heating load in urban areas decreased by 10.29% and the average annual cooling load increased by 7.48% compared with rural areas. $I_{UHI}$ is the main climatic factor affecting load variation. When $I_{UHI}$ increases by 1 °C, the urban heating load decreases by 4.01 kWh/m$^2$, and cooling load increases by 1.05 kWh/m$^2$. According to the research of Sun et al. [33], under the urban heat island effect, the energy consumption of building refrigeration is proportional to the size of the city. Hasid [34] studied the cooling load and maximum power consumption of buildings in different locations in the west of Athens, and the differences were 15–30% and 30–80%. Zindzi [35] studied the energy consumption of buildings around and inside Rome, which increased by 12% and 46%, respectively. According to statistics [36], nonresidential buildings use 40 percent more energy than residential buildings, especially service buildings.

Among all kinds of buildings, hospital buildings are particularly special in the aspect of air conditioning operation. Hospital air conditioning systems not only put forward strict requirements on indoor temperature and humidity but also need to ensure that the air is clean and sterile, in order to minimize the increase in energy consumption due to ventilation and ventilation times. Therefore, hospital building energy consumption simulation is more representative than other city building energy consumption simulations. Hospitals are public buildings, and the air conditioning energy consumption accounts for about 27% of total energy consumption [37]. In the total energy expenditure of hospital operation, the cost of the air conditioning system reaches more than 10% of the total cost of hospital operation. At the same time, the Energy Conservation Law, GB 50189-2015 "Energy Conservation Design Standard for Public Buildings", GB 19577-2015 "Cold Water Functional Efficiency Limit and Energy efficiency level" and other laws and regulations

and national mandatory standards have further promoted the development direction of air conditioning equipment energy conservation [38].

Building energy consumption cannot be calculated without the support of outdoor meteorological parameters [39]. However, in a rapidly warming climate, 30 years is too long a time for meteorological statistics to reflect climate trends and be effective in the design [40], planning, and decision making of climate services [41]. In the latest ASHRA [42], published in 2017, 25 years of meteorological data were used to determine the outdoor calculation parameters, which were conservatively shortened on a 30-year basis to take into account the effects of climate change by using the latest meteorological data to generate regularly updated outdoor calculation parameters. Argyris [43] used 20 years of hourly weather data from Athens, Kalogeria S A [44] used 7 years of hourly climate data from Nicosia, and Skeiner K [45] used 10 years of hourly weather data from Damascus. Ebrahim pours A et al. [46] used 14 years of hourly meteorological data for Bandar Abbas to generate locally applicable outdoor calculation parameters. A research team from Tianjin University of China conducted a lot of research. Based on the two basic principles of determining the statistical duration, Cao Xiang et al. determined the minimum statistical duration of outdoor dry bulb temperature calculation and the best mean value of meteorological elements according to the standard deviation method and the optimal climate normal (OCN) model and selected the statistical duration of outdoor dry bulb temperature calculation through comparison. Taking Tianjin as an example, the statistical duration of dry bulb temperature, calculated using the ASHRAE method and China method, is obtained [47].

At present, the selection of meteorological parameters in the field of air conditioning in China has some problems, such as incomplete parameter types and imperfect statistical methods [24]. Countries around the world began revising standards for design days in the 1930s. The current standard of ASHRAE in the United States considers the law of simultaneous occurrence of meteorological parameters, but using the average value of corresponding parameters as the combination index cannot accurately reflect the probability of occurrence of outdoor air conditioning points. Similar to the United States, the United Kingdom and Japan use the non-guaranteed rate form and do not classify by use. Since 1975, the selection of parameters has not changed substantially. In the case of simultaneous occurrence of meteorological parameters, the current method of selecting a single point value calculated according to the dry bulb temperature of outdoor air conditioning does not consider the hourly variation in wet bulb temperature and wind speed in the calculation of air conditioning load. ASHRAE uses simultaneous averaging of a single parameter and another parameter, and only considers partial simultaneity. All outdoor parameters occur at the same time. If each parameter is analyzed separately, the characteristics of parameter simultaneity will be ignored. Therefore, the simultaneous occurrence of humidity and wind speed should be considered in the selection of dry bulb temperature data, so as to ensure the accuracy and rationality of outdoor meteorological parameters. The rapid development of meteorological data and the lack of updating of research methods, coupled with the impact of climate change after the 1970s, have gradually derailed research results and engineering practice and gradually derailed international research. Therefore, in consideration of parameter coupling and data updates, the previous design day parameter selection method needs to be improved to meet the actual requirements of scientific research and engineering applications [48].

In China's "13th Five-Year Plan", building energy efficiency received more attention [38]. The proposal of a double carbon target further clarifies the development direction of the research on building energy consumption. With the deepening of a large number of related studies, meteorological data have been greatly supplemented and improved both in time and space, and their accuracy has been continuously improved. At the same time, the rapid development of artificial intelligence algorithms enables us to have more rapid and reliable means to study problems in the face of massive data, which provides the basis for the fine study of relevant parameters of air conditioning design. The Recurrent Neural

Network (RNN) [49] in machine learning algorithms is widely used to deal with time-series prediction because it has a recurrent neural network (RNN) that can better process sequential information, that is, the influence of the input between the front and back. Among them, Long Short-Term Memory network (LSTM), as a special RNN, can solve the common problem of long-term dependence in general recursive neural networks and has better performance in longer time series. Compared with other artificial intelligence algorithms, it is more suitable to study the selection of relevant meteorological parameters under climate change because it can retain the long-term dependence relationship caused by time characteristics to a greater extent [50].

In the "Design code for heating ventilation and air conditioning of civil buildings GB 50736-2012 [51], the hourly variation coefficient $\beta$ of the outdoor temperature is adopted by all places in a country, which not only lacks the timeliness of meteorological parameters but also does not meet the differences among regions, so it is in urgent need of updating and correction. In the context of climate change, Xiang Cao et al. [52] showed that 15 years was more reasonable than 30 years in the calculation of dry bulb temperature outside air conditioning in summer.

At present, some studies [53] focus on starting with air conditioning energy consumption and equipment selection under the condition that the capacity of cold source equipment is too large in most projects. By optimizing the annual mean unguaranteed value, the hourly temperature of the design day is reduced, and the equipment capacity is reduced as much as possible so as to reduce energy consumption. This experiment focused on exploring the regional diversity of $\beta$ value, and it corrected and predicted the variation trend of $\beta$ under the influence of climate change and the heat island effect, so as to provide a certain reference value for the selection of design days in the future.

Therefore, under the background of climate change, the change rule of meteorological data is constantly updated. The calculation parameters of previously determined outdoor air conditioning not only lack the real-time performance required by the research of meteorological and building energy consumption at the present stage but also lack the differences between different regions. By using more mature and advanced methods in the current research field, we improved the previous standardized calculation program, aiming to generate a set of summer air conditioning design day selection methods suitable for different regions in combination with the urban heat island effect, taking into account the comprehensive influence of temperature, humidity, and wind speed, and reflect the change law of outdoor air conditioning design day parameters in the future by using the prediction model. The building load model is used to compare the difference between the improved parameter values and the predicted values under the heat island effect and the previous standard values, which lays a foundation for the subsequent research.

## 2. Materials and Methods

By selecting hourly and daily meteorological data from Beijing and Shanghai, the CRITIC objective weight method was adopted to analyze the simultaneous occurrence of temperature, humidity, and wind speed data from Beijing and Shanghai in the past 15 years. The updated modified $\beta$ was generated by K-means clustering method, and then the LSTM neural network model was trained to predict the future trend of $\beta$. According to the meteorological data of Beijing and Shanghai in 15 years, the design day parameters suitable for local conditions are calculated. Finally, the intensity of urban heat island $\beta$ under the influence of heat island effect was obtained. Taking the air conditioning load of a representative building hospital as an example, the air conditioning load model was constructed, and the hourly cooling load per unit area in summer was calculated by updating and revising the design daily parameters, and the influence of urban heat island effect on the air conditioning load in summer was compared and analyzed.

The above data were calculated using the prediction temperature prediction algorithm based on approximate model matching [54]. The data came from the research groups related to Chinese architectural research, and the data source was real and reliable. In terms

of urban heat island, the urban heat island effect in Beijing is "strong at night and weak during the day" [55], while the fine structure of the urban heat island effect in Shanghai is "polycentric" [56]. Therefore, Chaoyang Station is selected as the center station, Beijing Station as the reference station to calculate the heat island intensity, Shanghai Jiading Station as the central station, and Xujiahui station as the reference station to calculate the heat island intensity, as shown in Figure 1. Among them, urban heat island intensity refers to the temperature difference between urban and rural areas caused by the urban heat island effect and other factors, as follows:

$$I_{UHI} = T_u - T_r \tag{1}$$

where $T_u$—city station air temperature (°C); $T_r$—reference station air temperature (°C).

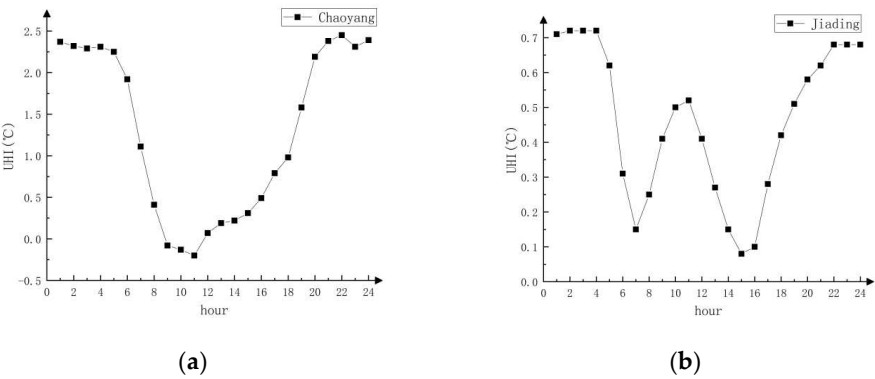

(**a**)                      (**b**)

**Figure 1.** Average heat island intensity of Chaoyang (**a**) and Jiading (**b**).

## 3. Experimental Data

### 3.1. Calculation Method

According to GB 50736-2012 "Design code for heating ventilation and air conditioning of civil buildings", Formulas (2) and (3) for calculating hourly outdoor air conditioning temperature in summer are shown in Table 2.

$$t_{sh} = t_{wp} + \beta \Delta t_r \tag{2}$$

$$\Delta t_r = \frac{t_{wg} - t_{wp}}{0.52} \tag{3}$$

where $t_{sh}$—hourly outdoor temperature (°C); $t_{wp}$—average daily temperature outside air conditioning in summer (°C); $\beta$—hourly variation coefficient of outdoor temperature; $\Delta t_r$—mean daily range outdoors in summer (°C); $t_{wg}$—dry bulb temperature outside air conditioning in summer (°C).

**Table 2.** Time-by-time variation coefficient of dry bulb temperature.

| Hour | 1 | 2 | 3 | 4 | 5 | 6 |
|------|------|------|------|------|------|------|
| $\beta$ | −0.35 | −0.38 | −0.42 | −0.45 | −0.47 | −0.41 |
| **Hour** | **7** | **8** | **9** | **10** | **11** | **12** |
| $\beta$ | −0.28 | −0.12 | 0.03 | 0.16 | 0.29 | 0.40 |
| **Hour** | **13** | **14** | **15** | **16** | **17** | **18** |
| $\beta$ | 0.48 | 0.52 | 0.51 | 0.43 | 0.39 | 0.28 |
| **Hour** | **19** | **20** | **21** | **22** | **23** | **24** |
| $\beta$ | 0.14 | 0.00 | −0.10 | −0.17 | −0.23 | −0.26 |

### 3.2. Hourly Change Coefficient of Outdoor Temperature β Modified

For cities in different climate zones across the country, only a set of hourly variation coefficient β of outdoor temperature is given in the standard, which cannot meet the fine calculation of air conditioning load. It is necessary to generate a set of β based on local meteorological conditions of the city to further study air conditioning load variation. The hourly change coefficient β of outdoor temperature was calculated based on the daily peak and daily mean of dry bulb temperature. The meteorological data were all summer dry bulb temperature values in July and August, and Formula (4) was as follows:

$$\begin{gathered} \beta_i = \frac{X_i - X_p}{X_{max} - X_p}, \beta_i \in [-1, 1], \\ i = 0, 1, 2, \ldots, 22, 23 \end{gathered} \tag{4}$$

where $\beta_i$—hourly trends of meteorological elements; i—moment; $X_i$—hour-by-hour value; $X_p$—average day; $X_{max}$—peak.

#### 3.2.1. Analysis of Simultaneity

For the simultaneous occurrence of meteorological parameters, the calculation of fresh air load at the present stage is selected according to the single point value design calculated by the dry bulb temperature calculated by outdoor air conditioning, without considering the hourly changes in wet bulb temperature and wind speed at the same time; each outdoor parameter occurs at the same time, and the simultaneous occurrence characteristics of parameters are ignored in the separate analysis of each parameter. Therefore, the simultaneous occurrence of humidity and wind speed should be taken into account when selecting the dry bulb temperature data, so as to ensure the accuracy and rationality of outdoor meteorological parameter selection.

Therefore, on the basis of the meteorological data of Beijing and Shanghai for 15 years previously selected, the β values of dry bulb temperature, wet bulb temperature, and wind speed were calculated using Formula (4). Since any two of the above three categories of meteorological elements are linear and continuous data, and any two categories of observations are paired and independent, showing normal distribution on the whole, satisfying the applicable conditions of the Pearson correlation coefficient. Therefore, the Pearson correlation coefficient is used to represent the degree of linear correlation among dry bulb temperature, wet bulb temperature, and wind speed. The calculation process is shown in Formula (5), and the results are shown in Figure 2. As can be seen from Figure 2, there is a strong correlation between dry bulb temperature and wet bulb temperature, a weak correlation between dry bulb temperature and wind speed, and a very weak correlation between wet bulb temperature and wind speed in Beijing and Shanghai.

$$r = \frac{1}{n-1} \sum_{i=1}^{n} \left( \frac{X_i - \overline{X}}{\sigma_X} \right) \left( \frac{Y_i - \overline{Y}}{\sigma_Y} \right) \tag{5}$$

Type: $X_i$, $Y_i$—dry bulb temperature β, wet bulb temperature β, and wind speed β, $\frac{X_i - \overline{X}}{\sigma_X}$ —the standard fraction of the sample $X_i$, $\frac{Y_i - \overline{Y}}{\sigma_Y}$ —the standard fraction of the sample $Y_i$, $\overline{X}$—sample mean of $X_i$, $\overline{Y}$—sample mean of $Y_i$, $\sigma_X$—sample standard deviation of $X_i$, $\sigma_Y$—sample standard deviation of $Y_i$.

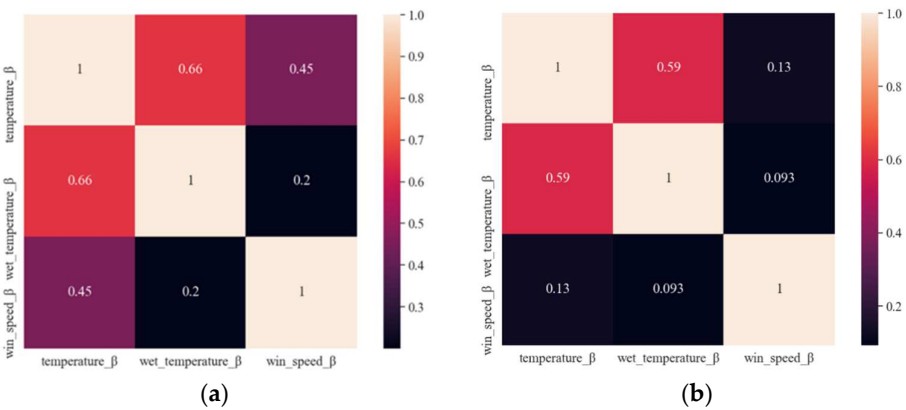

**Figure 2.** Correlation analysis of *β* between Beijing (**a**) and Shanghai (**b**).

### 3.2.2. CRITIC Empowerment

The CRITIC method comprehensively measures the objective weight of an indicator based on the comparative strength of an evaluation indicator and the conflict between the indicators. Considering both the variability in indicators and the correlation between indicators, the larger the number is, the less important it is. A scientific evaluation by fully utilizing the objective attributes of the data is a better objective weighting method than the entropy weight method and standard deviation method. Therefore, based on correlation analysis, to ensure the simultaneous occurrence of meteorological parameters, the CRITIC weight method is adopted to give corresponding weight values for dry bulb temperature, wet bulb temperature, and wind speed, respectively. The values can be used to characterize the simultaneous occurrence of meteorological parameters.

Firstly, relevant data were obtained—the hourly values of Beijing from 2003 to 2017 in summer (July and August). A total of $m(m = 23,808)$ subjects to be evaluated, dry bulb temperature $β$, wet bulb temperature $β$, and wind speed $β$ were $n(n = 3)$ evaluation indexes, forming the original data matrix $X$, where $X_{m,n}$ represents the value of the nth evaluation index of the mth sample, see Formula (6).

$$X = \begin{pmatrix} x_{1,1} & \cdots & x_{1,n} \\ \vdots & \ddots & \vdots \\ x_{m,1} & \cdots & x_{m,n} \end{pmatrix} \tag{6}$$

To eliminate the impact of dimensional inconsistencies on the evaluation results, it is necessary to conduct dimensionless processing of indicators so that all data can be measured by a unified standard. A larger value for the index is better, and the positive index is used, see Formula (7); if the value of the index used is smaller, the inverse index is used, see Formula (8).

$$x'_{ij} = \frac{x_{ij} - \min(x_j)}{\max(x_j) - \min(x_j)} \tag{7}$$

$$x'_{ij} = \frac{\max(x_j) - x_{ij}}{\max(x_j) - \min(x_j)} \tag{8}$$

In calculating the amount of information before, we first calculated the volatility $S_j$, as shown in Formula (9), where $x_j$ is the average for each indicator, and $S_j$ is the standard deviation of the first j an indicator.

$$S_j = \sqrt{\frac{\sum_{i=1}^{m}(x_{ij} - \overline{x}_j)^2}{n-1}} \tag{9}$$

The correlation matrix $R$ of the index should be used to calculate the conflict, see Formula (10).

$$= \frac{\sum_{j,k=1}^{n}\left(x_{ij} - \overline{x}_j\right)\left(x_{ik} - \overline{x}_k\right)}{\sqrt{\sum_{j=1}^{n}\left(x_{ij} - \overline{x}_j\right)^2 \sum_{k=1}^{n}\left(x_{ik} - \overline{x}_k\right)^2}} \tag{10}$$

Then, the conflict $A_j$ is calculated, as shown in Formula (11), which represents the correlation coefficient between the $i$th index and the $j$th index. The stronger the correlation with other indicators, the less conflict, and the greater the weight.

$$A_j = \sum_{i=1}^{n}\left(1 - r_{ij}\right) \tag{11}$$

To calculate the amount of information $C_j$, see Formula (12). The larger $C_j$ is, the greater the role of the jth evaluation index in the whole evaluation index system and the greater the corresponding weight.

$$C_j = S_j \times A_j \tag{12}$$

Finally, the weight $W_j$ was calculated, as shown in Formula (13). The calculation results are shown in Table 3, and $\beta_{syn}$ was synthesized by Formula (14).

$$W_j = \frac{C_j}{\sum_{j=1}^{n} C_j} \tag{13}$$

$$\beta_{syn} = W_{jtem}\beta_{tem} + W_{jwet}\beta_{wet} + W_{jws}\beta_{ws} \tag{14}$$

**Table 3.** Calculation results of Beijing $\beta$ weight.

|  |  | $\beta_{tem}$ | $\beta_{wet}$ | $\beta_{ws}$ |
|---|---|---|---|---|
| $S_j$ | Beijing | 0.128813 | 0.047748 | 0.061137 |
| | Shanghai | 0.078371 | 0.066816 | 0.078753 |
| $A_j$ | Beijing | 1.790672 | 1.538196 | 2.653585 |
| | Shanghai | 1.538942 | 1.505506 | 2.219937 |
| $C_j$ | Beijing | 0.230662 | 0.073446 | 0.162232 |
| | Shanghai | 0.120609 | 0.100592 | 0.174828 |
| $W_j$ | Beijing | 0.494622 | 0.157495 | 0.347883 |
| | Shanghai | 0.304546 | 0.254001 | 0.441452 |

### 3.2.3. K-Means Clustering

Since the change law of $\beta_{syn}$ in different periods is different, corresponding to the change trend of 24 h is different, so the selected $\beta_{syn}$ is classified, and the K-means clustering analysis method is used to classify $\beta_{syn}$ into four categories according to the change law. The classification results and the proportion of each category are shown in Figures 3–5. Among them, Class A, Class B, Class C, and Class D refer to the four $\beta_{syn}$ change types formed after K-means clustering. Each curve in the picture represents a change in $\beta_{syn}$ and is distinguished by a different color.

Among them, the average value of $\beta_{tem}$ in Beijing and Shanghai is taken as the average value of the category with the largest proportion; that is, Beijing chooses the average value of $\beta_{syn}$ of class A, and Shanghai chooses the average value of $\beta_{syn}$ of class A, and then the corresponding $\beta_{tem}$ is calculated according to the formula. The results are shown in Tables 4 and 5.

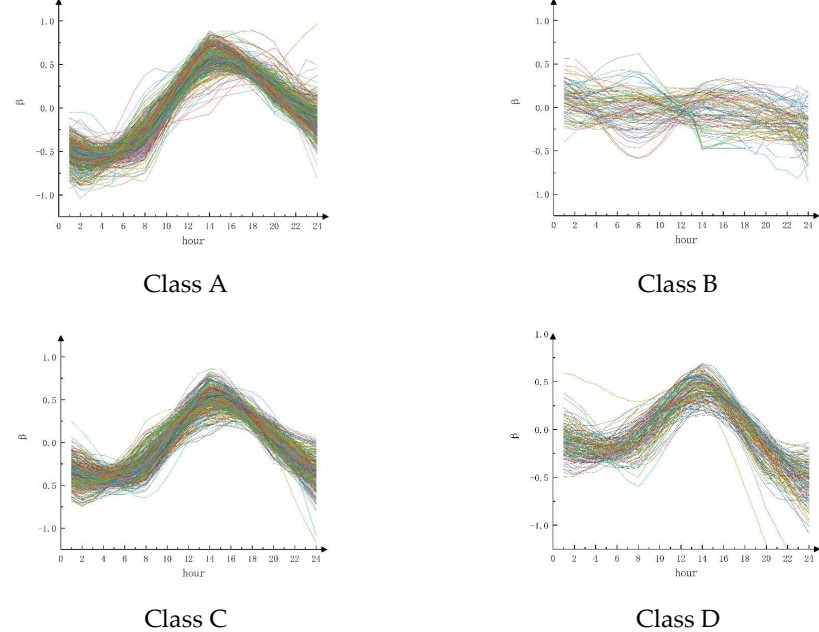

**Figure 3.** Beijing $\beta_{syn}$ classification.

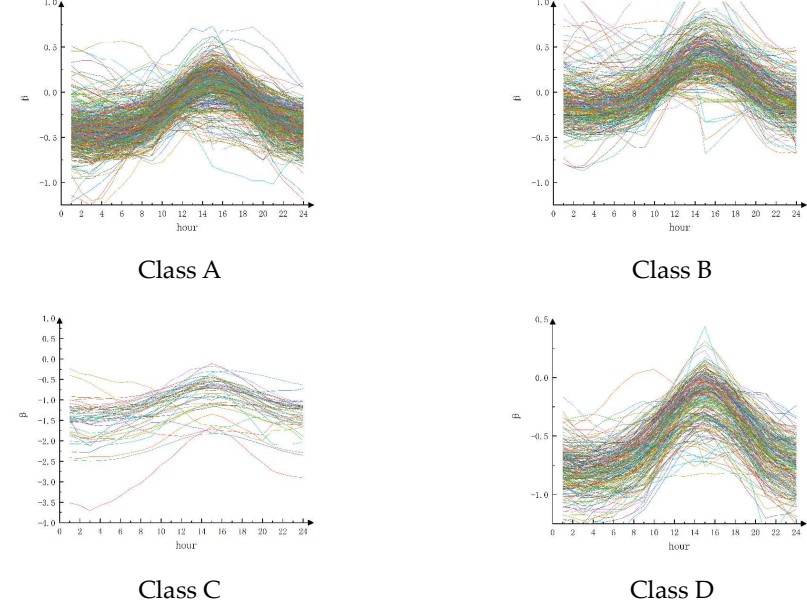

**Figure 4.** Shanghai $\beta_{syn}$ classification.

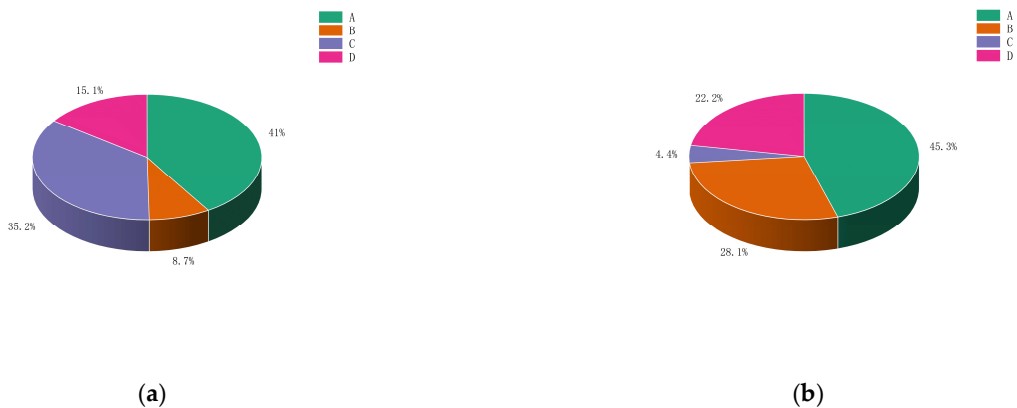

**Figure 5.** Proportions of various types in Beijing (**a**) and Shanghai (**b**).

**Table 4.** Hourly variation coefficient of dry bulb temperature in Beijing.

| Hour | 1 | 2 | 3 | 4 | 5 | 6 |
|---|---|---|---|---|---|---|
| Beijing $\beta_{tem}$ | −0.67 | −0.74 | −0.78 | −0.79 | −0.76 | −0.69 |
| **Hour** | 7 | 8 | 9 | 10 | 11 | 12 |
| Beijing $\beta_{tem}$ | −0.58 | −0.42 | −0.21 | 0.04 | 0.29 | 0.52 |
| **Hour** | 13 | 14 | 15 | 16 | 17 | 18 |
| Beijing $\beta_{tem}$ | 0.71 | 0.83 | 0.87 | 0.84 | 0.76 | 0.63 |
| **Hour** | 19 | 20 | 21 | 22 | 23 | 24 |
| Beijing $\beta_{tem}$ | 0.48 | 0.31 | 0.16 | 0 | −0.13 | −0.26 |

**Table 5.** Hourly variation coefficient of dry bulb temperature in Shanghai.

| Hour | 1 | 2 | 3 | 4 | 5 | 6 |
|---|---|---|---|---|---|---|
| Shanghai $\beta_{tem}$ | −0.13 | −0.16 | −0.17 | −0.18 | −0.18 | −0.17 |
| **Hour** | 7 | 8 | 9 | 10 | 11 | 12 |
| Shanghai $\beta_{tem}$ | −0.13 | −0.05 | 0.07 | 0.23 | 0.43 | 0.62 |
| **Hour** | 13 | 14 | 15 | 16 | 17 | 18 |
| Shanghai $\beta_{tem}$ | 0.80 | 0.93 | 0.99 | 0.96 | 0.86 | 0.71 |
| **Hour** | 19 | 20 | 21 | 22 | 23 | 24 |
| Shanghai $\beta_{tem}$ | 0.53 | 0.34 | 0.18 | 0.06 | −0.04 | −0.10 |

### 3.2.4. LSTM Long Short-Term Memory Network Model Construction and Prediction

LSTM was designed to solve the problem of long-term dependence, which is common in general recursive neural networks; that is, it can effectively transmit information in a long time series without causing previously useful information to be ignored. Based on the K-means clustering analysis, the LSTM algorithm was used to build a prediction model. Thus, 24 h was taken as the length of a set of sample data, and 80% training set and 20% of the prediction set data were set to train and verify the model. Combined with the weight calculation results in Tables 4 and 5, the future, $\beta_{syn}$, $\beta_{wet}$, and $\beta_{ws}$ were predicted, and the $\beta_{tem}$ was calculated via Formula (14). Since the training set accounts for 80% and the test set accounts for 20%, in the prediction model based on the data for 15 years, we believe that the prediction of data changes in the next 3 years has strong confidence. The LSTM cell structure and prediction model structure are shown in Figure 6, where the red circle represents the operation on the vector (para operation), the yellow rectangle represents a neural network layer, and the characters represent the activation function used by the neural network layer.

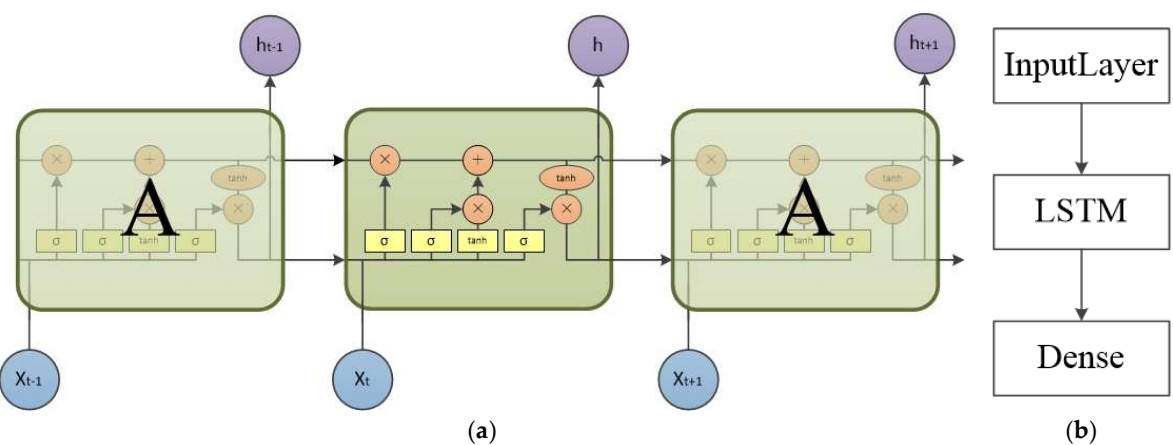

**Figure 6.** LSTM cell structure (**a**) and prediction model structure (**b**).

Among them, the results of $\beta_{syn}$, $\beta_{wet}$, and $\beta_{ws}$ are shown in Figure 7. The fitting degree of $\beta_{syn}$, $\beta_{wet}$, and $\beta_{ws}$ is shown in Figure 8. The closer the value of $r^2$ is to 1, the better the prediction effect will be. The corresponding $r^2$ values of Beijing and Shanghai are shown in Table 6.

**Figure 7.** Distribution of $\beta_{syn}$, $\beta_{wet}$, and $\beta_{ws}$ in Beijing (**a**) and Shanghai (**b**).

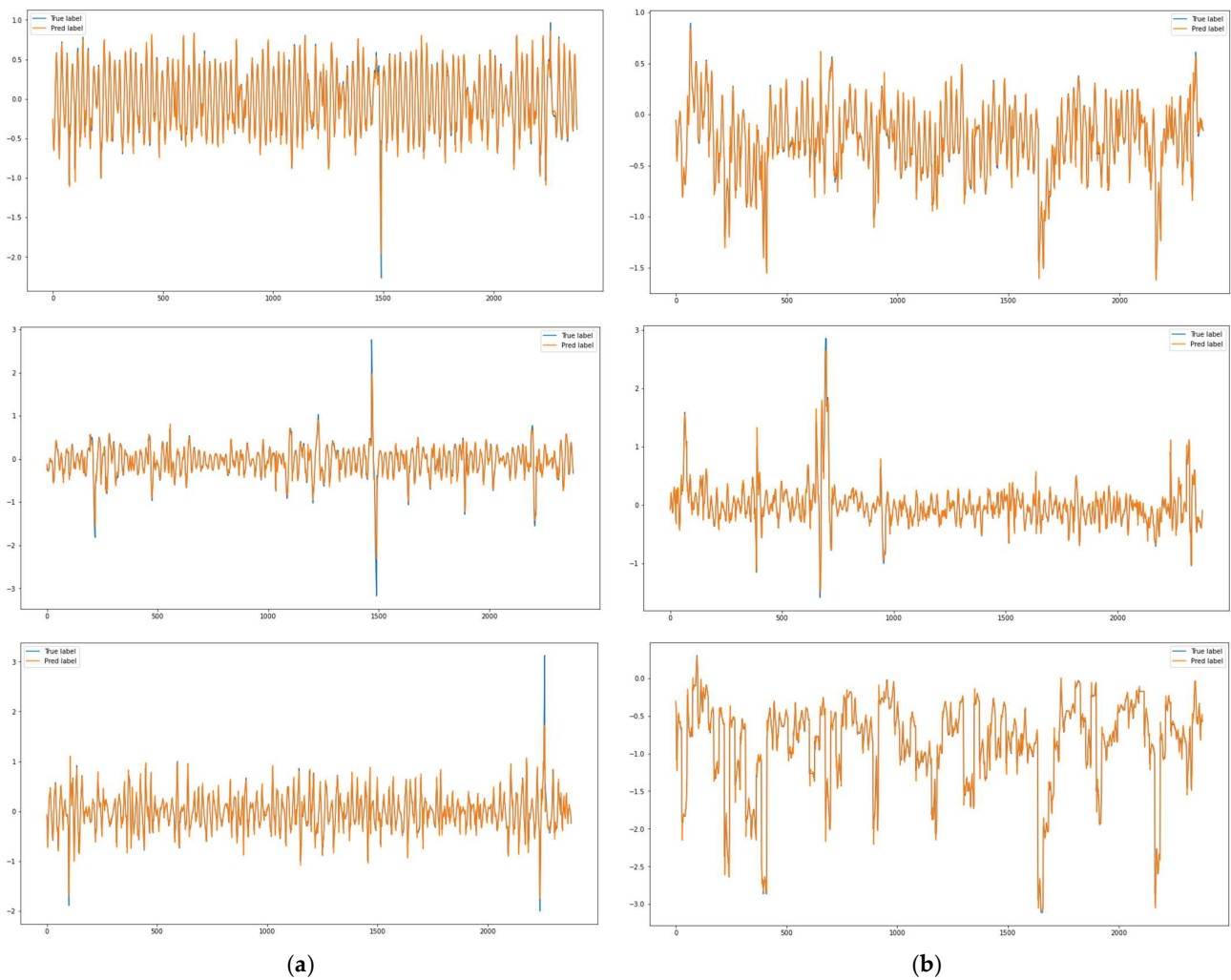

**Figure 8.** Fitting degree of $\beta_{syn}$, $\beta_{wet}$, and $\beta_{ws}$ in Beijing (**a**) and Shanghai (**b**).

**Table 6.** $r^2$ values for Beijing and Shanghai.

| $.r^2$ | $\beta_{syn}$ | $\beta_{wet}$ | $\beta_{ws}$ |
|---|---|---|---|
| Beijing | 0.96 | 0.89 | 0.89 |
| Shanghai | 0.93 | 0.90 | 0.92 |

Figure 9 shows the predicted results of $\beta_{syn}$, $\beta_{wet}$, and $\beta_{ws}$. In the figure, blue is the actual value, and green is the predicted value.

Tables 7 and 8 are used to calculate $\hat{\beta}_{tem}$ in Beijing and Shanghai based on the equation.

A comparison between the revised value based on K-means clustering and the predicted value of LSTM model and the standard value is shown in Figure 10.

According to the comparison results, compared with $\beta$ in the specification, the amplitude of $\beta$ obtained by the K-means method is larger in the Beijing area. In the data for the Shanghai group, except for the data at 9:00 am, $\beta$ obtained by the K-means method is higher than $\beta$ in the specification. According to the prediction results of LSTM, the variation trend and amplitude of $\beta$ predicted in the Beijing group data were roughly the same as those in the specification, and the peak appeared 2 h earlier, and the data from 7:00 to 15:00 were higher than those in the specification, while the predicted value of LSTM in the Shanghai group data was higher than that of K-means.

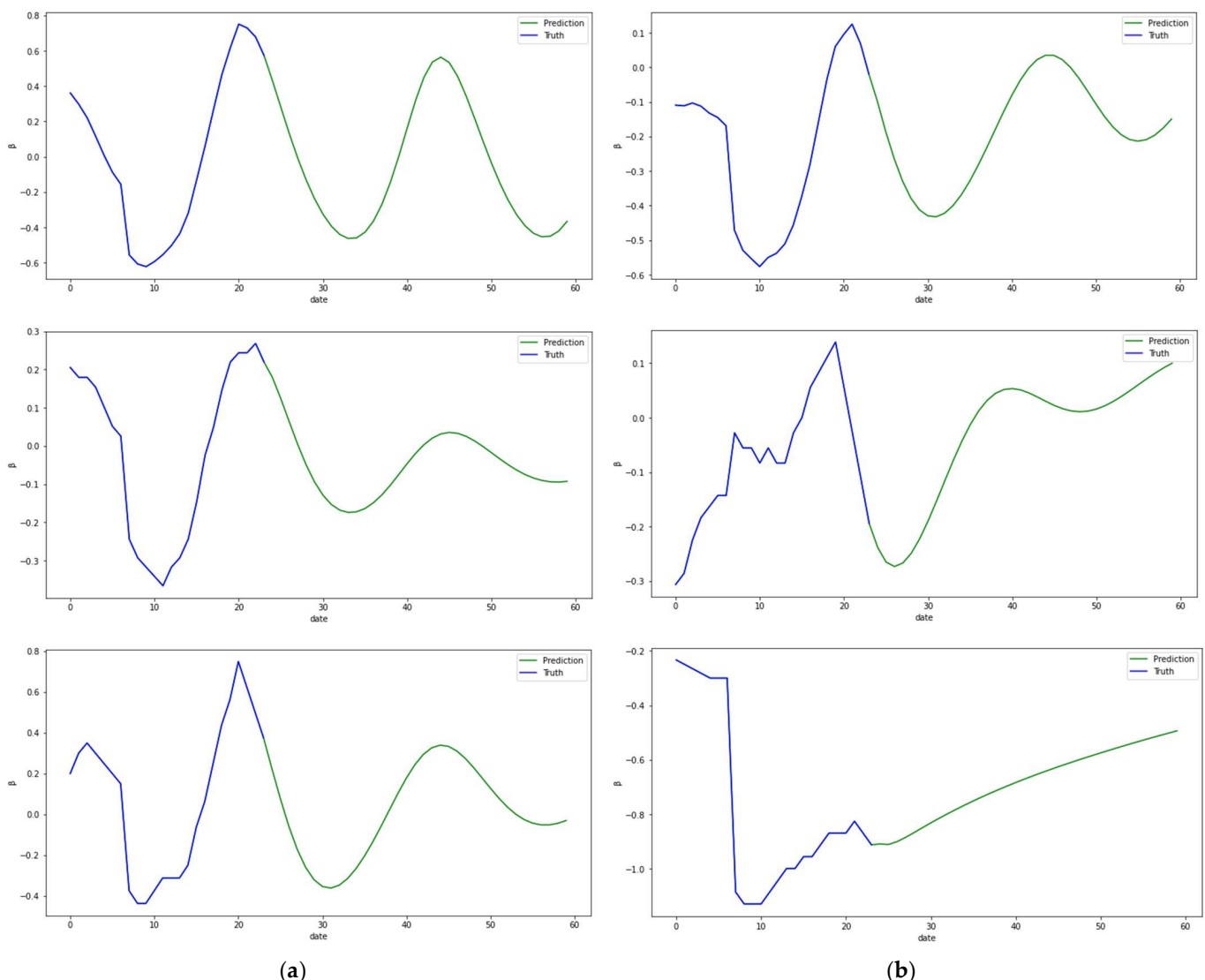

(**a**)　　　　　　　　　　　　　　　　　　　　(**b**)

**Figure 9.** Predicted results of $\beta_{syn}$, $\beta_{wet}$, and $\beta_{ws}$ in Beijing (**a**) and Shanghai (**b**).

**Table 7.** Predicted hourly variation coefficient of dry bulb temperature in Beijing.

| Hour | 1 | 2 | 3 | 4 | 5 | 6 |
|---|---|---|---|---|---|---|
| Beijing $\hat{\beta}_{tem}$ | −0.25 | −0.34 | −0.40 | −0.42 | −0.41 | −0.37 |
| **Hour** | **7** | **8** | **9** | **10** | **11** | **12** |
| Beijing $\hat{\beta}_{tem}$ | −0.31 | −0.24 | −0.17 | −0.05 | 0.07 | 0.21 |
| **Hour** | **13** | **14** | **15** | **16** | **17** | **18** |
| Beijing $\hat{\beta}_{tem}$ | 0.34 | 0.44 | 0.48 | 0.47 | 0.41 | 0.32 |
| **Hour** | **19** | **20** | **21** | **22** | **23** | **24** |
| Beijing $\hat{\beta}_{tem}$ | 0.22 | 0.10 | −0.03 | −0.15 | −0.26 | −0.35 |

**Table 8.** Predicted hourly variation coefficient of dry bulb temperature in Shanghai.

| Hour | 1 | 2 | 3 | 4 | 5 | 6 |
|---|---|---|---|---|---|---|
| Shanghai $\hat{\beta}_{tem}$ | 0.17 | 0.08 | 0.03 | 0.03 | 0.06 | 0.13 |
| Hour | 7 | 8 | 9 | 10 | 11 | 12 |
| Shanghai $\hat{\beta}_{tem}$ | 0.23 | 0.36 | 0.51 | 0.67 | 0.82 | 0.96 |
| Hour | 13 | 14 | 15 | 16 | 17 | 18 |
| Shanghai $\hat{\beta}_{tem}$ | 1.08 | 1.60 | 1.20 | 1.21 | 1.17 | 1.10 |
| Hour | 19 | 20 | 21 | 22 | 23 | 24 |
| Shanghai $\hat{\beta}_{tem}$ | 1.00 | 0.88 | 0.75 | 0.62 | 0.50 | 0.39 |

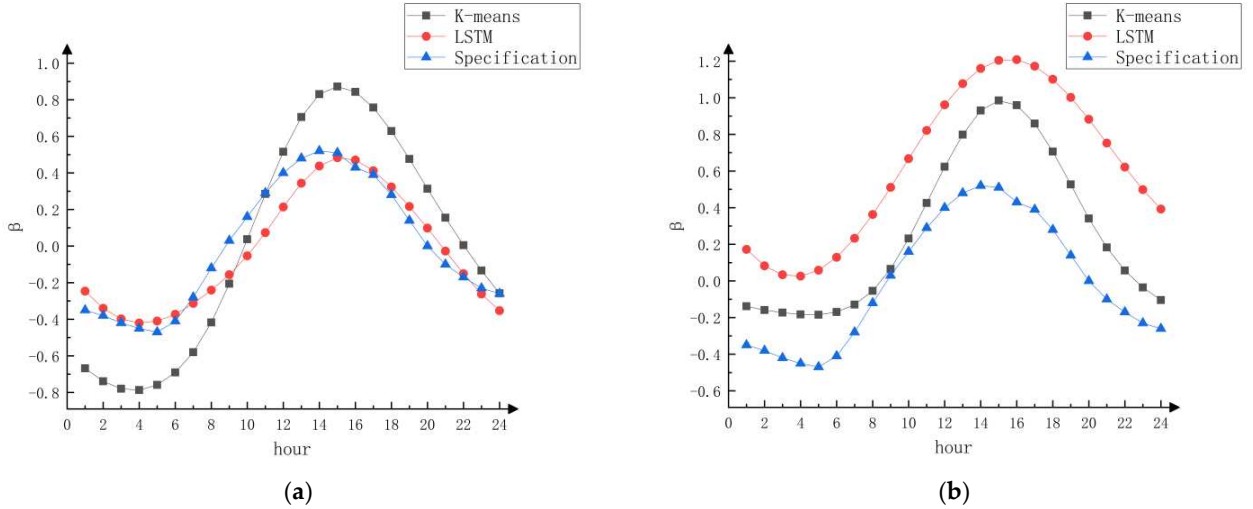

**Figure 10.** Predicted value $\hat{\beta}_{tem}$ Comparison of $\beta_{syn}$, $\beta_{wet}$, and $\beta_{ws}$ in Beijing (**a**) and Shanghai (**b**).

*3.3. Summer Air Conditioning Calculated Temperature and Summer Air Conditioning Calculated Daily Average Temperature Correction*

According to the "Guidelines for Indoor and Outdoor Parameters for HVAC Design of Civil Buildings" [56], the design day is mainly selected by dry bulb temperature, and the calculated temperature of air conditioning in summer and the daily average temperature of air conditioning in summer are selected (see Table 9).

**Table 9.** Selection method by dry bulb temperature.

| **Select by dry bulb temperature** | Air conditioning temperature calculation in summer | Annual average does not guarantee the average dry bulb temperature of 10 h, 50 h, and 100 h |
|---|---|---|
| | Calculate the average daily temperature of air conditioning in summer | Annual average does not guarantee the daily average temperature of 1 d, 5 d, and 10 d |

When selecting the calculated temperature of summer air conditioning, the data of the 151st, 751st, and 1501st groups were selected in descending order according to the hourly value of dry bulb temperature, and then all the data within the design value of ±0.5 °C were selected to take the mean value, corresponding to the calculated temperature of summer air conditioning that does not guarantee the annual average of 10 h, 50 h, and 100 h. When selecting summer air conditioning to calculate the daily average temperature, the data of the 16th, 76th, and 151st groups were selected in descending order according

to the daily average of dry bulb temperature, which corresponded to the daily average temperature of summer air conditioning without guaranteeing 1 d, 5 d, and 10 d of annual average, respectively. Tables 10 and 11 show the selection results.

**Table 10.** Selection results in Beijing.

| Beijing | Cumulative Average Does Not Guarantee 10 h | Cumulative Average Does Not Guarantee 50 h | Cumulative Average Does Not Guarantee 100 h |
|---|---|---|---|
| Air conditioning temperature calculation in summer | 35.4 °C | 33.6 °C | 32.3 °C |
| **Beijing** | **Averaging over years does not guarantee 1 d** | **Averaging over years does not guarantee 5 d** | **Averaging over years does not guarantee 10 d** |
| Calculate the average daily temperature of air conditioning in summer | 31.3 °C | 29.9 °C | 29.1 °C |

**Table 11.** Selection results in Shanghai region.

| Shanghai | Cumulative Average Does Not Guarantee 10 h | Cumulative Average Does Not Guarantee 50 h | Cumulative Average Does Not Guarantee 100 h |
|---|---|---|---|
| Air conditioning temperature calculation in summer | 38.1 °C | 35.9 °C | 34.8 °C |
| **Shanghai** | **Averaging over years does not guarantee 1 d** | **Averaging over years does not guarantee 5 d** | **Averaging over years does not guarantee 10 d** |
| Calculate the average daily temperature of air conditioning in summer | 33.1 °C | 31.4 °C | 30.7 °C |

In Beijing and Shanghai, the calculation of summer air conditioning temperature and the calculation of daily average temperature in summer air conditioning are, respectively, selected as the result of annual average without a guarantee of 50 h and the result of annual average without a guarantee of 5 d.

*3.4. Design Day Revision*

According to the previously selected hourly change coefficient of dry bulb temperature $\beta$, calculated temperature $t_{wg}$ of summer air conditioning, and calculated daily average temperature $t_{wp}$ of summer air conditioning, the formula is substituted to calculate the design daily hourly dry bulb temperature $t_{sh}$ of Beijing and Shanghai, as shown in Tables 12 and 13.

**Table 12.** Beijing district design day.

| Hour | 1 | 2 | 3 | 4 | 5 | 6 |
|---|---|---|---|---|---|---|
| °C | 25.20 | 24.69 | 24.41 | 24.36 | 24.56 | 25.04 |
| **Hour** | **7** | **8** | **9** | **10** | **11** | **12** |
| °C | 25.81 | 26.96 | 28.45 | 30.16 | 31.91 | 33.53 |
| **Hour** | **13** | **14** | **15** | **16** | **17** | **18** |
| °C | 34.86 | 35.75 | 36.04 | 35.83 | 35.23 | 34.33 |
| **Hour** | **19** | **20** | **21** | **22** | **23** | **24** |
| °C | 33.25 | 32.11 | 30.99 | 29.93 | 28.96 | 28.08 |

**Table 13.** Shanghai district design day.

| Hour | 1 | 2 | 3 | 4 | 5 | 6 |
|------|-----|-----|-----|-----|-----|-----|
| °C | 30.20 | 30.01 | 29.89 | 29.81 | 29.79 | 29.92 |
| Hour | 7 | 8 | 9 | 10 | 11 | 12 |
| °C | 30.27 | 30.93 | 31.97 | 33.42 | 35.11 | 36.83 |
| Hour | 13 | 14 | 15 | 16 | 17 | 18 |
| °C | 38.36 | 39.50 | 39.98 | 39.76 | 38.88 | 37.55 |
| Hour | 19 | 20 | 21 | 22 | 23 | 24 |
| °C | 35.99 | 34.37 | 32.99 | 31.90 | 31.09 | 30.49 |

A comparison between the generated design day and the actual hourly temperature change is shown in Figure 11. As can be seen from the figure, the design day in the specification cannot effectively represent the hourly outdoor temperature change trend in summer, and the generated design day in Beijing and Shanghai has some changes in the value and occurrence time of the maximum temperature based on the original basis, and the predicted design day can represent the hourly temperature change in the future. In line with the effects of climate change, they can be better applied to the actual project requirements.

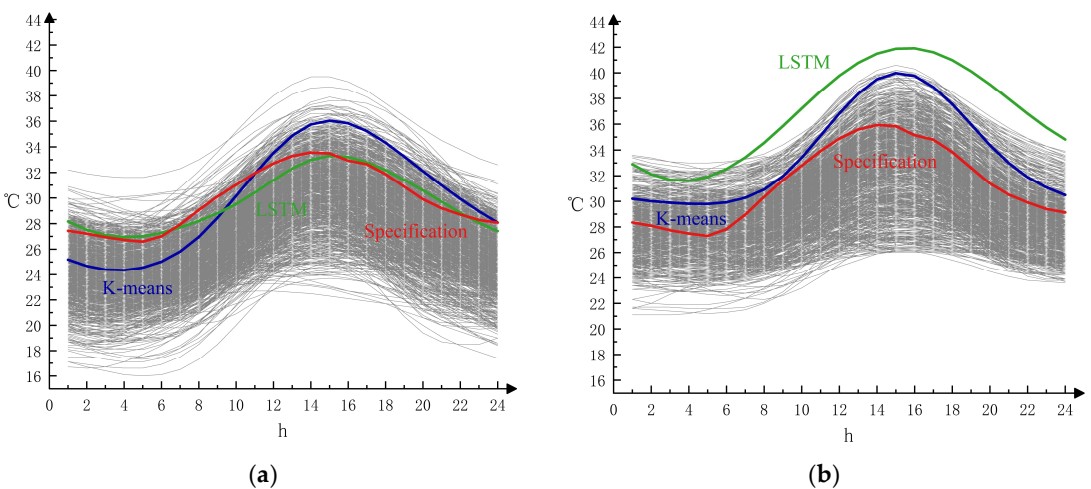

**Figure 11.** Comparison between Beijing (**a**) and Shanghai (**b**) design day and real value.

*3.5. Design Day Correction in the UHI Zone*

According to the modified hourly dry bulb temperature $t_{sh}$ and the urban heat island intensity $I_{UHI}$ of the corresponding region, the hourly dry bulb temperature $t_{sh-UHI}$ under the influence of the urban heat island effect is calculated as the design day in the region. Meanwhile, the hourly variation coefficient $\beta_{UHI}$ of dry bulb temperature under the influence of the urban heat island effect in this region is calculated according to the formula, which is taken as the $\beta$ of this region, as shown in Tables 14 and 15. Comparing the modified design days and regional design days under the UHI effect with the design days formed by $\beta$ in the original specification, the results are shown in Figure 12.

**Table 14.** Hourly variation coefficient of dry bulb temperature in Chaoyang.

| Hour | 1 | 2 | 3 | 4 | 5 | 6 |
|---|---|---|---|---|---|---|
| Chaoyang $\beta_{UHI}$ | −0.33 | −0.41 | −0.45 | −0.46 | −0.44 | −0.42 |
| **Hour** | **7** | **8** | **9** | **10** | **11** | **12** |
| Chaoyang $\beta_{UHI}$ | −0.42 | −0.36 | −0.22 | 0.02 | 0.26 | 0.53 |
| **Hour** | **13** | **14** | **15** | **16** | **17** | **18** |
| Chaoyang $\beta_{UHI}$ | 0.73 | 0.86 | 0.92 | 0.91 | 0.87 | 0.77 |
| **Hour** | **19** | **20** | **21** | **22** | **23** | **24** |
| Chaoyang $\beta_{UHI}$ | 0.70 | 0.62 | 0.49 | 0.35 | 0.19 | 0.08 |

**Table 15.** Hourly variation coefficient of dry bulb temperature in Jiading.

| Hour | 1 | 2 | 3 | 4 | 5 | 6 |
|---|---|---|---|---|---|---|
| Jiading $\beta_{UHI}$ | −0.06 | −0.08 | −0.09 | −0.1 | −0.11 | −0.13 |
| **Hour** | **7** | **8** | **9** | **10** | **11** | **12** |
| Jiading $\beta_{UHI}$ | −0.11 | −0.03 | 0.11 | 0.29 | 0.49 | 0.67 |
| **Hour** | **13** | **14** | **15** | **16** | **17** | **18** |
| Jiading $\beta_{UHI}$ | 0.83 | 0.95 | 0.99 | 0.97 | 0.89 | 0.75 |
| **Hour** | **19** | **20** | **21** | **22** | **23** | **24** |
| Jiading $\beta_{UHI}$ | 0.59 | 0.41 | 0.25 | 0.14 | 0.04 | −0.03 |

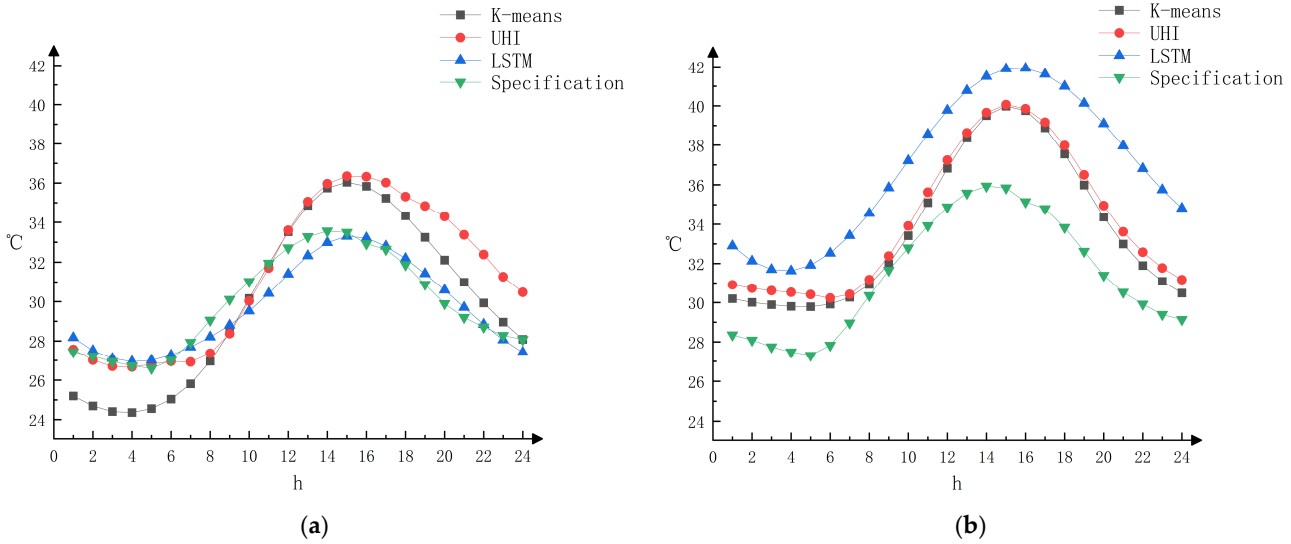

**Figure 12.** Design day comparison in Beijing (**a**) and Shanghai (**b**).

As can be seen from Figures 3–11, the full-day maximum temperature in Beijing in summer is one hour later than the norm and appears at 15:00, with a temperature rise of about 3 °C. It is expected to remain at 15:00 in the future, with the temperature roughly the same as the normal value. The temperature in Chaoyang District increases significantly from 15:00 to 7:00 in the morning of the next day, and the temperature from 8:00 to 14:00 is roughly the same as that in Beijing. The maximum temperature in summer in Shanghai was delayed 1 h later than the norm and appeared at 15:00, with a temperature rise of about 4 °C. In the future, it is expected to be delayed 1 h to 16:00, with a temperature rise of about 6 °C on the basis of the normal value. The temperature in Jiading District of Shanghai increases from 17:00 to 6:00 in the morning of the next day, and the temperature from 7:00 to 16:00 is roughly the same as that in Shanghai.

## 4. Modeling of Air Conditioning Load in Hospital Buildings

*4.1. Hospital Building Overview*

DeST is a software platform developed by Tsinghua University for the simulation of the built environment and HVAC system, which provides practical and reliable software tools for simulation prediction and performance evaluation of the built environment. The architectural model selected in this study is the outpatient department building and the inpatient department building of an actual hospital building. The architectural model is shown in Figure 13. Among them, the outpatient department has four floors, the first layer is 5.1 m high, the second to third layers are 3.8 m high, and the area is the same. The first floor contains a Chinese and Western pharmacy, internal medicine, gynecology, surgery, and other outpatient departments as well as a relatively independent emergency hall. On the second floor, there are departments for surgery, outpatient transfusion, and medical equipment. In the medical technology area, there are laboratory departments, blood testing centers, endoscopy centers, and functional tests. On the third floor, a hepatobiliary surgery department, physical examination center, and reserved departments are located. Medical technology is taken as examination department, with B-ultrasound, electrocardiogram, speculum center, and other examination rooms. On the fourth floor, there is a blood bank center, ICU, surgery center, expert outpatient clinic, dermatology, ophthalmology, otolaryngology, stomatology, and other examination rooms. The outpatient floor length is about 138 m, the width is about 105 m, and the overall construction area is about 57,900 m$^2$. The inpatient department has a total of 12 floors, the height is 3.8 m, and each floor is the same size. There are 12 rooms for three people, 3 rooms for two people, 1 room for five people, and only 1 room for the doctor's office, the director's office, the professor's office, and other rooms. The length of the inpatient floor is about 80 m, the width is about 19 m, and the total construction area is about 37,700 m$^2$.

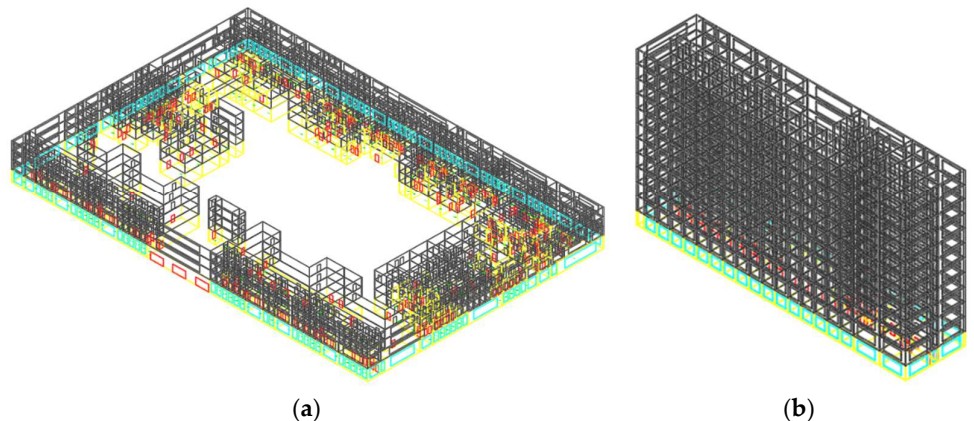

|  (a)  |  (b)  |

**Figure 13.** Building model of outpatient building (**a**) and inpatient building (**b**).

According to the General code for energy efficiency and renewable energy application in buildings GB55015-2021 [57]. and the Design standard for energy efficiency of public buildings GB50189-2015 [58], the thermal parameters of the envelope are set, as shown in Tables 16 and 17.

**Table 16.** Parameters of building envelope in Beijing.

| Structure | Material | Thickness (mm) | Density (kg/m$^3$) | Thermal Conductivity (W·m$^{-1}$·K$^{-1}$) | Specific Heat Capacity (J·kg$^{-1}$·K$^{-1}$) | Total Thermal Conductivity (W·m$^{-1}$·K$^{-1}$) | Total Heat Transfer Coefficient (W·m$^{-2}$·K$^{-1}$) |
|---|---|---|---|---|---|---|---|
| Exterior wall | Cement mortar | 20 | 1800 | 0.93 | 837 | 1.898 | 0.486 |
| | Reinforced concrete | 200 | 2500 | 1.628 | 837 | | |
| | Polystyrene foam | 50 | 100 | 0.047 | 1380 | | |
| | Expanded perlite | 85 | 120 | 0.058 | 670 | | |
| | Cement free fiberboard | 20 | 250 | 0.076 | 2512 | | |
| | Cement mortar | 20 | 1800 | 0.93 | 837 | | |
| Interior wall | Soot aerated concrete | 100 | 800 | 0.349 | 837 | 0.33 | 1.788 |
| | Cement mortar | 20 | 1800 | 0.93 | 837 | | |
| Floor | Cement mortar | 25 | 1800 | 0.93 | 837 | 0.098 | 3.054 |
| | Reinforced concrete | 80 | 2500 | 1.628 | 837 | | |
| | Cement mortar | 20 | 1800 | 0.93 | 837 | | |
| Roof | Cement mortar | 20 | 1800 | 0.93 | 837 | 1.047 | 0.812 |
| | Porous concrete | 200 | 600 | 0.209 | 837 | | |
| | Reinforced concrete | 130 | 2500 | 1.628 | 837 | | |
| | Cement mortar | 20 | 1800 | 0.93 | 837 | | |
| Floor level | Cement mortar | 20 | 1800 | 0.93 | 837 | 0.978 | - |
| | Porous concrete | 200 | 600 | 0.209 | 837 | | |

**Table 17.** Parameters of building envelope in Shanghai.

| Structure | Material | Thickness (mm) | Density (kg/m$^3$) | Thermal Conductivity (W·m$^{-1}$·K$^{-1}$) | Specific Heat Capacity (J·kg$^{-1}$·K$^{-1}$) | Total Thermal Conductivity (W·m$^{-1}$·K$^{-1}$) | Total Heat Transfer Coefficient (W·m$^{-2}$·K$^{-1}$) |
|---|---|---|---|---|---|---|---|
| Exterior wall | Reinforced concrete | 200 | 2500 | 1.628 | 837 | 1.45 | 0.622 |
| | Pure drywall | 10 | 1100 | 0.407 | 837 | | |
| | Polystyrene foam | 60 | 100 | 0.047 | 1380 | | |
| | Pure drywall | 8 | 1100 | 0.407 | 837 | | |
| Interior wall | Cement mortar | 20 | 1800 | 0.93 | 837 | 0.33 | 1.788 |
| | Soot aerated concrete | 100 | 800 | 0.349 | 837 | | |
| | Cement mortar | 20 | 1800 | 0.93 | 837 | | |
| Floor | Cement mortar | 25 | 1800 | 0.93 | 837 | 0.098 | 3.054 |
| | Reinforced concrete | 80 | 2500 | 1.628 | 837 | | |
| | Cement mortar | 20 | 1800 | 0.93 | 837 | | |
| Roof | Cement mortar | 20 | 1800 | 0.93 | 837 | 1.047 | 0.812 |
| | Porous concrete | 200 | 600 | 0.209 | 837 | | |
| | Reinforced concrete | 130 | 2500 | 1.628 | 837 | | |
| | Cement mortar | 20 | 1800 | 0.93 | 837 | | |
| Floor level | Cement mortar | 20 | 1800 | 0.93 | 837 | 0.047 | - |
| | Gravel | 200 | 600 | 0.209 | 837 | | |

### 4.2. Indoor Calculation Conditions

According to the requirements in GB51039-2014, the temperature of ordinary air conditioning should be above 20 °C in winter and not higher than 27 °C in summer. There should be a fresh air supply and exhaust, and the system scale should not be too large, as shown in Table 18. According to the requirements in JGJ/T 499-2018 "Calculation Standard for Green Performance of Civil Buildings [Description of the Accompanying Provisions]", the room zoning parameters of medical and health buildings are shown in Table 19. According to JGJ/T 499-2018 Green Performance Calculation Standard for Civil Buildings (Supplementary Provisions), the simulation calculation operation parameters of the air conditioning system are shown in Figure 14, where t is the temperature set for different types of rooms in summer. When the fresh air is running, 1 means the fresh air is flowing and 0 means it is not.

**Table 18.** Calculate the air conditioner temperature in each area.

| Name of Occupancy | Calculation of the Temperature (°C) |
|---|---|
| inpatient ward | 20~24 |
| consultation room, examination, treatment room | 18~24 |
| bathroom and lavatory for patients | 22~26 |
| general operating room, delivery room | 20~24 |
| office and activity room | 18~20 |
| no movable room (such as pharmaceutical depot) | ≥ 10 |

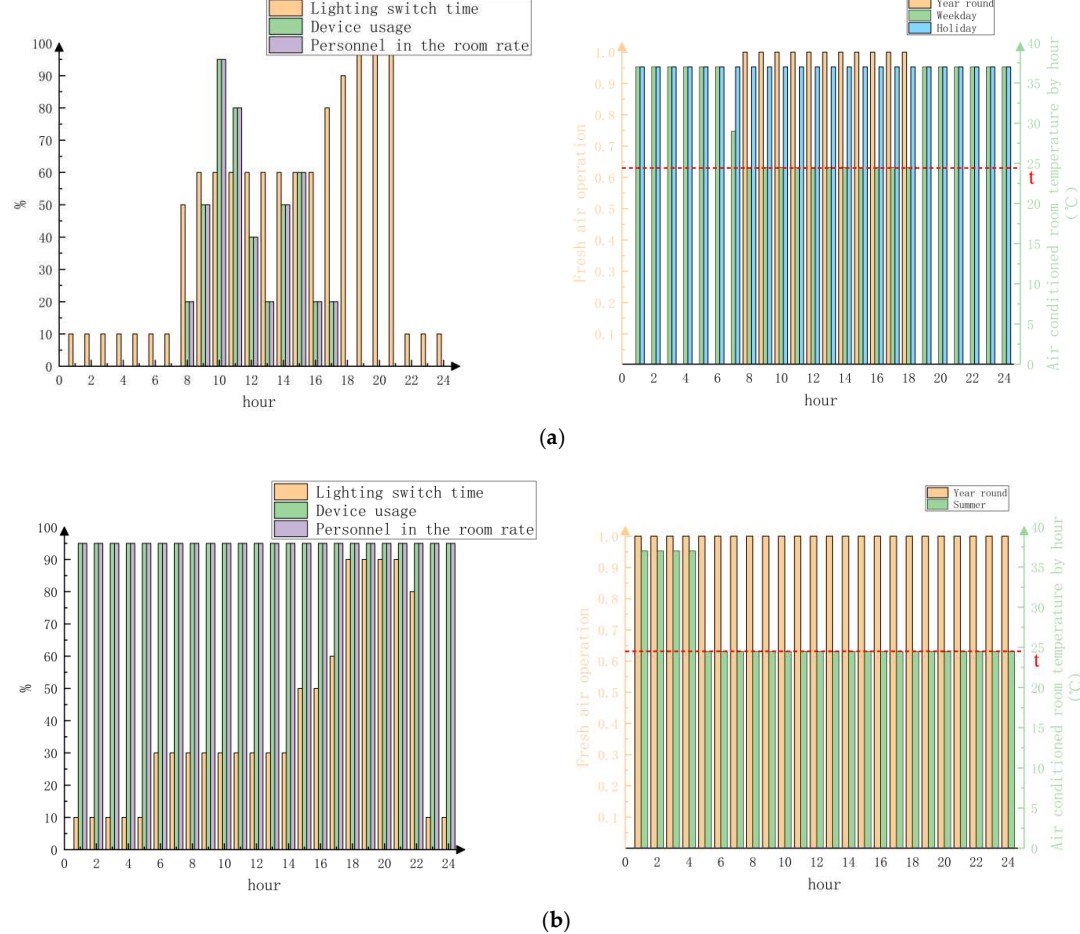

**Figure 14.** The operating parameters of the air conditioning system in the outpatient department (**a**) and inpatient department (**b**) were calculated by simulation.

**Table 19.** Room partitioning parameters for healthcare buildings.

| The Name of the Partition | Illumination Power Density (W/m$^2$) | Equipment Power Density (W/m$^2$) | Personnel Density (m$^2$/Person) | Personnel Heat Dissipation (W/Person) | The New Air Volume | | Set the Temperature of the Room in Summer (°C) |
|---|---|---|---|---|---|---|---|
| | | | | | (m$^3$/h·Person) | (Times/h) | |
| The pharmacy | 17 | 20 | 10 | 134 | — | 2 | 26 |
| Between devices | 6 | 20 | — | 134 | — | — | — |
| Office | 9 | 20 | 6 | 134 | 30 | — | 26 |
| Warehouse | 5 | 20 | — | 134 | — | — | 28 |
| Treatment room | 9 | 20 | 6 | 134 | — | 2 | 16 |
| Transfusion room | 9 | 20 | 20.5 | 108 | — | 2 | 26 |
| Waiting Registered hall | 6 | 20 | 4 | 134 | 60 | — | 27 |
| Intensive care | 9 | 20 | 4 | 181 | — | 2 | 26 |
| The emergency room | 9 | 20 | 4 | 181 | — | 2 | 26 |
| Laboratory | 15 | 20 | 10 | 134 | — | 2 | 26 |
| The operating room | 25 | 20 | 10 | 235 | 60 | — | 26 |
| The meeting room | 9 | 20 | 2.5 | 134 | 14 | — | 26 |
| B-mode ultrasonography | 9 | 20 | 10 | 134 | 30 | — | 26 |
| Ward | 5 | 20 | 5 | 108 | — | 2 | 26 |
| The restaurant | 9 | 20 | 2.5 | 134 | 30 | — | 26 |
| Intensive ICU | 9 | 20 | 8 | 181 | 60 | — | 26 |
| Computer room | 6 | 20 | — | — | — | — | — |
| The nurse station | 9 | 20 | 8 | 181 | 30 | — | 26 |
| Toilet | 6 | 20 | 20 | 134 | — | — | 28 |
| Stair | 5 | 20 | — | — | — | — | — |
| The corridor | 5 | 20 | 8 | 108 | 30 | — | 26 |

## 5. Experimental Results and Analysis

*Experimental Results*

The experimental results are shown in Figure 15. For the same type of hospital building, the total cooling load in summer in Shanghai is higher than that in Beijing. For different types of hospital buildings, the total cooling load of inpatient buildings in summer is higher than that of outpatient buildings. As for the variation in cooling load in summer, in the Beijing area, for the outpatient building of the hospital, the building cooling load modified by K-means is reduced by 0.69% compared with before, and the building cooling load in the Chaoyang area is increased by 12.12% compared with before under the heat island effect. The LSTM model predicted that the total cooling load of the building was reduced by 1.35% compared to the previous one. For hospital inpatient buildings, the building refrigeration modified by the K-means method increased by 0.27% compared to before, the building cooling load in the Chaoyang area increased by 7.13% compared to before under the heat island effect, and the building cooling load predicted by the LSTM model decreased by 0.93% compared to before. In Shanghai, for the outpatient building of the hospital, the building cooling load modified by the K-means method increased by 12.61% compared to before, the building cooling load in the Jiading area increased by 15.51% under the heat island effect, and the total building cooling load predicted by LSTM model increased by 29.75% compared to before. For hospital inpatient buildings, the building cooling load modified by the K-means method increased by 6.71% compared with before, the building cooling load in the Jiading area increased by 8.09% compared with before under the heat island effect, and the total building cooling load predicted by the LSTM model increased by 16.07% compared with before.

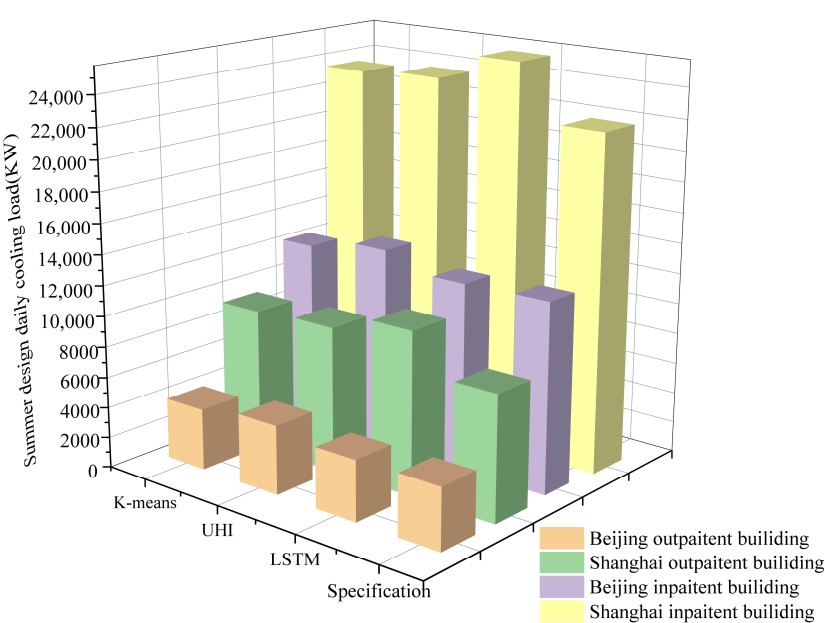

**Figure 15.** Summer design daily cooling load.

As can be seen from Figures 16 and 17, for different types of hospital buildings, the operating conditions of the outpatient building are different from those of the inpatient building. The total cooling load of inpatient buildings in summer is higher than that of outpatient buildings. Among them, a small amount of convective heat dissipation in lighting heat dissipation can be directly converted into an air conditioning cooling load, while massive radiation heat dissipation forms a lagging air conditioning cooling load, which results in the overall lag of the impact of lighting switch time on improvements in air conditioning cooling load in the building. Combined with Figure 14, the cooling load of the unit area of the outpatient building in summer decreased slightly and then increased due to the fluctuation in the personnel room rate and equipment utilization rate from 12:00 to 14:00. In the period from 20:00 to 22:00, a decrease in the personnel occupancy rate and equipment utilization rate from 16:00 to 20:00 has a greater impact than an increase in lighting switch time, so the energy consumption of air conditioning and refrigeration still maintains a downward trend from 16:00 to 20:00, and from 20:00 to 21:00, the influence brought by the accumulated lighting switch time rise before began to appear, so that the cooling load of air conditioning increased, and 21:00 to 22:00 switching lighting time dropped sharply, resulting in the cooling load of air conditioning. The cooling load of the unit area of the inpatient building in summer is from 16:00 to 20:00, and the outdoor temperature continues to decrease, leading to the cooling load of the air conditioning. From 22:00 to 23:00, as the lighting switch time had been rising slowly before, the cumulative impact gradually appeared, resulting in a small rebound in the air conditioning cooling load.

Figures 18 and 19 show the cooling load of different types of hospital buildings in summer. The top four outpatient buildings in summer cooling load in the Beijing area are archives warehouse, pharmacy, B-mode ultrasonography, and infusion, with loads of 357.71 kW·h/m$^2$, 142.12 kW·h/m$^2$, 138.11 kW·h/m$^2$, and 137.15 kW·h/m$^2$, respectively. The top four outpatient buildings in summer cooling load in the Shanghai area are archives warehouse, infusion, waiting room, and conference room, with loads of 496.86 kW·h/m$^2$, 320.68 kW·h/m$^2$, 308.18 kW·h/m$^2$, and 160.27 kW·h/m$^2$, respectively. In summer, the cooling load of the ward and nurse station is the largest for the air conditioning load of inpatient buildings. The load in the Beijing area is 998.44 kW·h/m$^2$ and 117.39 kW·h/m$^2$, respectively. The loads in Shanghai are 1905.03 kW·h/m$^2$ and 180.32 kW·h/m$^2$, respectively. The air conditioning cooling load of hospital buildings in the Shanghai area is generally higher than that in the Beijing area in summer.

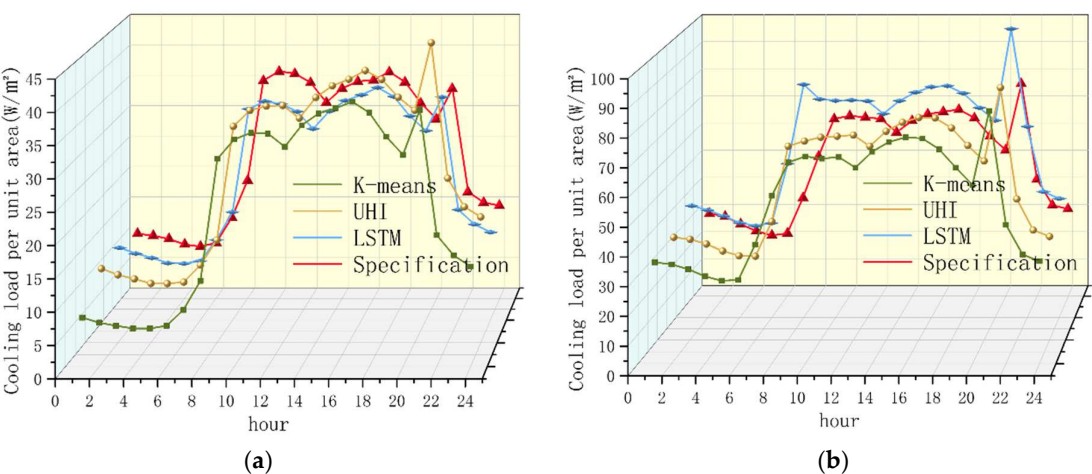

(**a**)                                                                                              (**b**)

**Figure 16.** The unit area of outpatient buildings in Beijing (**a**) and Shanghai (**b**) is subjected to hourly cooling load.

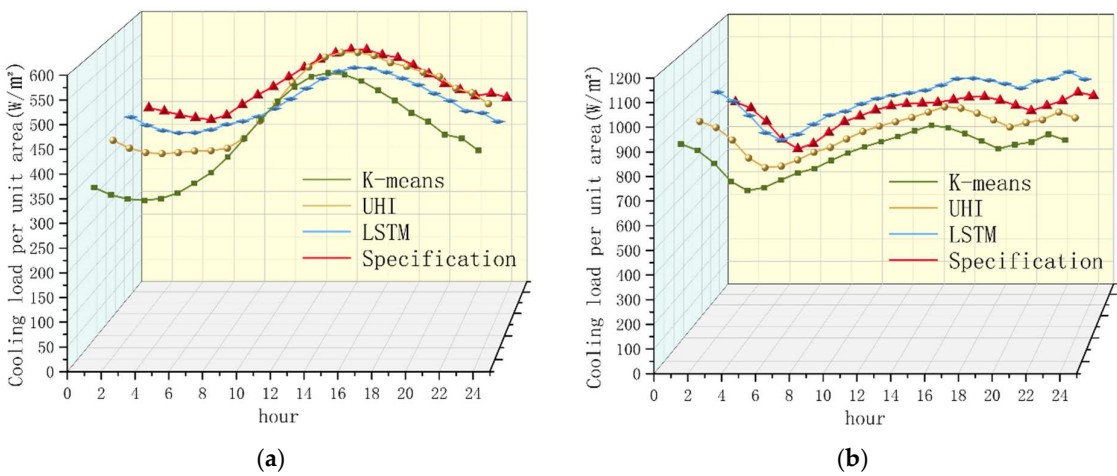

(**a**)                                                                                              (**b**)

**Figure 17.** The unit area of inpatient buildings in Beijing (**a**) and Shanghai (**b**) is subjected to hourly cooling load.

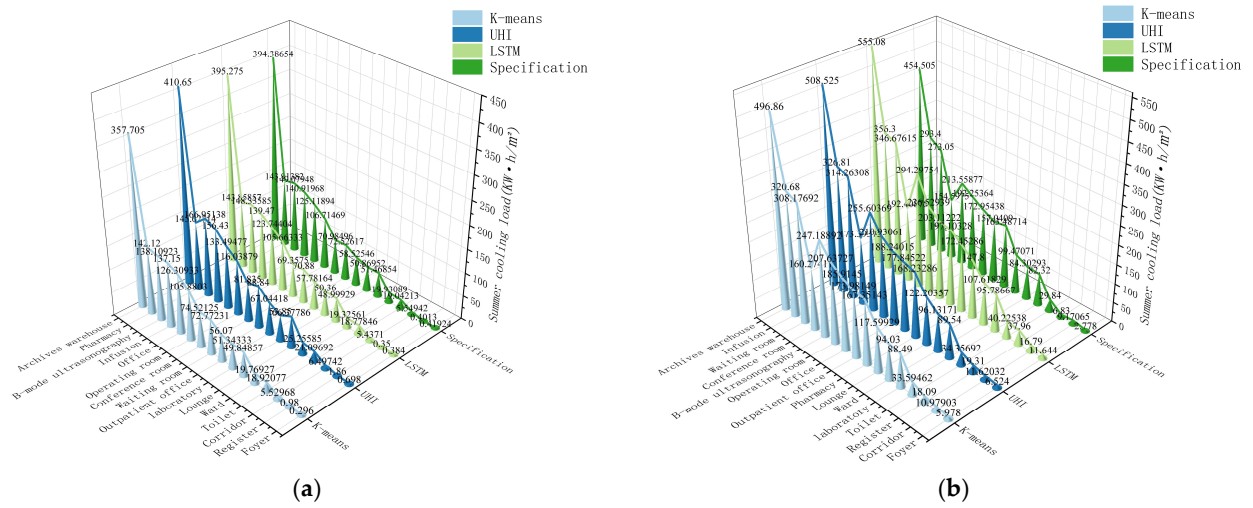

(**a**)                                                                                              (**b**)

**Figure 18.** Statistics of summer cooling load in Beijing (**a**) and Shanghai (**b**) outpatient buildings.

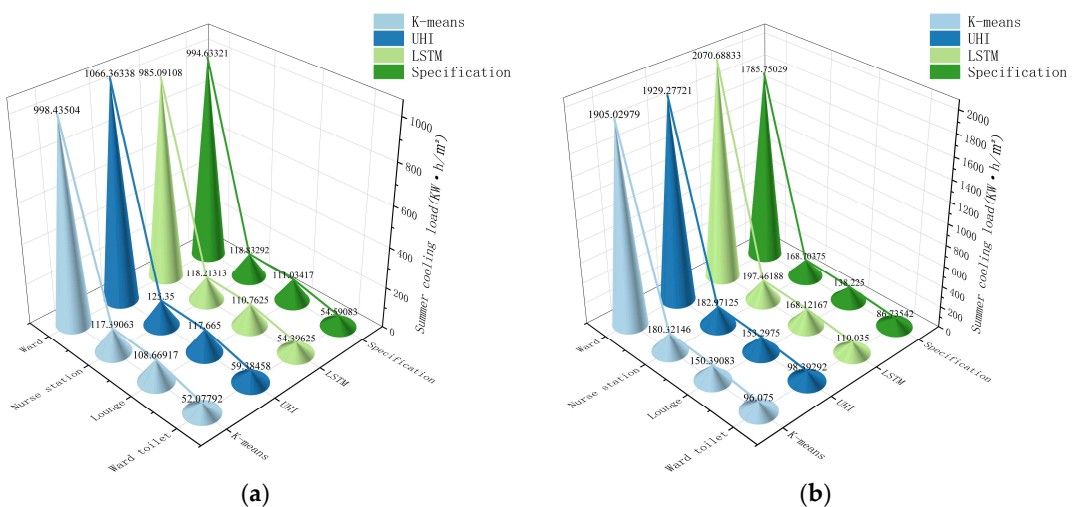

**Figure 19.** Statistics of summer cooling load in Beijing (**a**) and Shanghai (**b**) inpatient buildings.

## 6. Discussion

Considering the influence of climate change, an appropriate meteorological statistical period (15 years) was selected, and based on the hourly data and daily mean data of dry bulb temperature, wet bulb temperature, and wind speed in Beijing and Shanghai over 15 years, the hourly variation coefficient $\beta$ of dry bulb temperature in "Design Code for Heating, Ventilation and Air Regulation of Civil Buildings" GB 50736-2012 was modified. Through the Pearson correlation analysis and the CRITIC weight, the coupling between different meteorological elements was considered. The K-means clustering method was adopted to classify $\beta$, and the $\beta$ of the class with the largest proportion was selected to calculate the mean value. The $\beta$ meeting the local meteorological conditions was generated, and the LSTM neural network was used to predict the change in $\beta$ in a future period. At the same time, the calculated temperature of summer air conditioning and the calculated daily average temperature of summer air conditioning were re-selected to generate the revised local design days. Then, the average urban heat island intensity in different regions (Beijing selected the Chaoyang area, Shanghai selected the Jiading area) was used to calculate the design days of Chaoyang and Jiading under the influence of the urban heat island effect and the corresponding $\beta$. Finally, taking the hospital building as an example, DeST software was used to model the outpatient building and inpatient building, and the changes in hourly cooling load and hourly cooling load per unit area on the modified design day were analyzed under the influence of the urban heat island effect. Through the analysis of the summer cooling load of different functional rooms, the characteristics of the hospital building in summer cooling load were reflected. It can be seen from the experimental results that, compared with before the correction, the hourly cooling load of the unit area of the hospital building changes significantly in summer:

(1) Compared with the design day in the specification, the revised design day is more in line with the actual local meteorological conditions and has obvious regional differences. The maximum temperature in Beijing was one hour later than the norm and appeared at 15:00, with a temperature rise of about 2 °C. In the future, the temperature is expected to stabilize at 15:00, with the temperature falling back to the normal value. In Shanghai, the maximum temperature of the whole day was 1 h later than the norm and appeared at 15:00, with a temperature rise of about 4 °C. In the future, it is expected to continue to be 1 h to 16:00, with a temperature rise of about 6 °C based on the normal value.

(2) The influence time and intensity of the heat island effect on the summer cooling load in different cities are different. The load in the Chaoyang District of Beijing increased significantly from 15:00 to 7:00 am of the next day in summer, and the load from 8:00

to 14:00 was roughly the same as before the correction. The standard load ratio of outpatient buildings in hospitals will decrease by 0.69%, increase by 12.12% under the influence of the heat island effect, and decrease by 1.35% in the future; the standard load ratio of inpatient buildings will increase by 0.27%, increase by 7.13% under the influence of the heat island effect, and decrease by 0.93% in the future. It shows that climate change and the heat island effect have an obvious influence on the increase in the air conditioning cooling load in Beijing in the short term, but in the long term, the cooling load will fall back to the previous level. In the Jiading District of Shanghai, the load increased significantly from 17:00 to 6:00 in the morning of the next day, and the load from 7:00 to 16:00 was roughly the same as before the correction. The standard load ratio of outpatient buildings in hospitals increased by 12.61%, 15.51% under the influence of the heat island effect, and 29.75% in the future; the standard load ratio of inpatient buildings increased by 6.71%, 8.09% under the influence of the heat island effect, and 16.07% in the future. It shows that climate change and the heat island effect have an obvious influence on the increase in the air conditioning cooling load in Shanghai in the short term, and the cooling load will continue to increase in the long term.

(3) The response intensity of different types of hospital buildings to the influence of summer cooling load is different: in Beijing hospital, the top four outpatient buildings in the summer cooling load are archives warehouse, pharmacy, B-mode ultrasonography, and infusion, while in Shanghai hospital, the top four outpatient buildings in summer cooling load are archives warehouse, infusion, waiting room, and conference room. For hospital inpatient buildings in Beijing and Shanghai, the cooling load of wards is the largest in summer, followed by the nurse station. The total cooling load of other rooms in summer is less than that of the nurse station. The total cooling load of inpatient buildings in summer is generally higher than that of outpatient buildings, which has greater energy saving potential.

## 7. Conclusions

In view of the existing problems of outdoor air conditioning design parameters, we proposed a new calculation program for outdoor air conditioning design day parameter standardization based on K-means clustering and an LSTM neural network algorithm. The selection period of appropriate meteorological data (15 years) solves the problem that the meteorological data for the previous 30 years are not applicable to the selection of air conditioning parameters under the influence of climate change. The meteorological and urban heat island data of Beijing and Shanghai were used to solve the problem of a lack of regional difference in the previous method, and the influence of the urban heat island effect was taken into account. By using the method of CRITIC weighting, the influence of temperature, humidity, and wind speed was comprehensively analyzed, and the problem of insufficient consideration of the simultaneous occurrence of meteorological parameters in the previous method was solved. A prediction model based on a neural network was built to predict the variation trend of outdoor air conditioning design daily parameters in the next 10 years, providing a basis for future related research. The air conditioning load model of a hospital was established, and the load was compared with the previous load to prove the load variation under the new standardized calculation program. In general, it solves the problems that the previous research failed to take into account, and successfully introduces the artificial intelligence algorithm into the new standardized computing program. In the next study, relevant data for a wider range of cities should be selected to focus on the change in air conditioning design parameters in different climate zones, so as to explore the influence of the climate change process on outdoor design day parameters of air conditioning in different regions.

**Author Contributions:** H.L. and L.S.: improvements in research methods; L.S.: modeling, calculation and manuscript writing. C.L. assisted in data analysis, code debugging and experimental data processing. J.L. helped edit the manuscript and modified the English grammar of the manuscript. C.L. and J.L. put forward useful suggestions and participated in the preparation of the manuscript. H.L. reviewed the manuscript and oversaw the project. The results are discussed and analyzed by all authors. All authors have read and agreed to the published version of the manuscript.

**Funding:** This study was supported by the National Natural Science Foundation of China. The project was supported by Research on Spatio-temporal Expansion of Meteorological Year for Building Energy Conservation (No. 52278124).

**Institutional Review Board Statement:** Not applicable.

**Informed Consent Statement:** Not applicable.

**Data Availability Statement:** Not applicable.

**Conflicts of Interest:** The authors declare no conflict of interest. The funders played no role in the design of the research; in the collection, analysis, or interpretation of data; in the writing of a manuscript; or in the decision to publish the results.

## Nomenclature

| Symbol | Meaning |
|---|---|
| $I_{UHI}$ | Urban heat island intensity |
| $\beta$ | Hourly variation coefficient of outdoor temperature |
| temperature_$\beta$, $\beta_{tem}$ | Temperature corresponds to $\beta$ |
| wet_temperature_$\beta$, $\beta_{wet}$ | Wet bulb temperature corresponds to $\beta$ |
| win_speed_$\beta$, $\beta_{ws}$ | Wind speed corresponds to $\beta$ |
| $\beta_{syn}$ | $\beta$ was synthesized by temperature, wet bulb temperature and wind speed |
| $S_j$ | Volatility |
| R | Index of the correlation matrix |
| $A_j$ | Conflict |
| $C_j$ | Amount of information |
| $W_j$ | Weight |
| $\hat{\beta}_{tem}$ | The predicted value of temperature $\beta$ |
| $\hat{\beta}_{wet}$ | The predicted value of wet bulb temperature $\beta$ |
| $\hat{\beta}_{ws}$ | The predicted value of wind speed $\beta$ |
| $\hat{\beta}_{syn}$ | The predicted value of synthesized $\beta$ |
| $\beta_{UHI}$ | $\beta$ under urban heat island effect |

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
