# Peer review of "Research on Air-Conditioning Cooling Load Correction and Its Application Based on Clustering and LSTM Algorithm"

_applsci, doi:10.3390/app13085151_

Round 1
Reviewer 1 Report
See the attachment

Author Response
Thank you for your time and all your suggestions on our articles.
Please see the attachment.

Reviewer 2 Report
The study aims to investigate how climate change and urban heat island effect affect the energy consumption of buildings in urban heat island areas. The authors used meteorological data and heat island intensity data to predict hourly change coefficient beta of dry bulb temperature using the CRITIC objective weighting method and K-means clustering analysis, and then established the air conditioning load model of a hospital by correcting and predicting the cooling load of air conditioning in summer using the LSTM algorithm. The results show that the new design days parameters are more consistent with the actual local meteorological conditions, and there is a difference in the cooling load of outpatient and inpatient hospital buildings in Beijing and Shanghai under the influence of heat island effect. The study suggests that the cooling load of outpatient buildings in Beijing will decrease in the future, while that in Shanghai will increase. On the other hand, the cooling load of inpatient buildings in Beijing and Shanghai will increase in the future.
Overall, the paper introduces good contributions and several assessment results; however, the presentation is very week.
1) The presentation of this paper needs a major revision. Some of the figures are not well presented and are not complete. Please revise your paper carefully before submitting it. I recommend for the authors to write their articles with Latex top avoid such kinds of issues.
2) The Introduction is too long better to split into part: (I) Introduction, and (II) Related works.
3) The contribution of the paper is not well presented. This should be reformulated in the Introduction and the authors can summarize the main contributions using bullet points.
4) The first equation in page 9 does not has a number. Please fix this issue and review all the equations to make sure that all of them are numbered.
5) Please number the Table and Figures without section numbers. For instance, Table 1, 2 or 3 (no need for Table 1-1, 1-2, or 1-3)
6) A Conclusion should be added to the paper after the Discussion, where the important findings, limitation of the proposed method and future directions should be discussed.
Author Response

(The authors gave the same response as above.)

Round 2
Reviewer 1 Report
After analyzing the revised text sent to me, I find that the article has been improved and my comments in the previous review have been sufficiently taken into account.
Thanks to the authors for responding to my comments.
Technical note:
Table 16. Information on individual elements should be separated by horizontal lines, as in Table 17.
Reviewer 2 Report
The authors have addressed all my comments; I have ano further suggestions.